# Nature-inspired remodeling of (aza)indoles to *meta*-aminoaryl nicotinates for late-stage conjugation of vitamin B₃ to (hetero)arylamines

Begur Vasanthkumar Varun [1,2], Kannan Vaithegi [1,2], Sihyeong Yi [1] & Seung Bum Park [1✉]

Despite the availability of numerous routes to substituted nicotinates based on the Bohlmann–Rahtz pyridine synthesis, the existing methods have several limitations, such as the inevitable *ortho*-substitutions and the inability to conjugate vitamin B₃ to other pharmaceutical agents. Inspired by the biosynthesis of nicotinic acid (a form of vitamin B₃) from tryptophan, we herein report the development of a strategy for the synthesis of *meta*-aminoaryl nicotinates from 3-formyl(aza)indoles. Our strategy is mechanistically different from the reported routes and involves the transformation of (aza)indole scaffolds into substituted *meta*-aminobiaryl scaffolds via Aldol-type addition and intramolecular cyclization followed by C–N bond cleavage and re-aromatization. Unlike previous synthetic routes, this biomimetic method utilizes propiolates as enamine precursors and thus allows access to *ortho*-unsubstituted nicotinates. In addition, the synthetic feasibility toward the halo-/boronic ester-substituted aminobiaryls clearly differentiates the present strategy from other cross-coupling strategies. Most importantly, our method enables the late-stage conjugation of bioactive (hetero)arylamines with nicotinates and nicotinamides and allows access to the previously unexplored chemical space for biomedical research.

---

[1] CRI Center for Chemical Proteomics, Department of Chemistry, Seoul National University, Seoul 08826, Republic of Korea. [2]These authors contributed equally: Begur Vasanthkumar Varun, Kannan Vaithegi. ✉email: sbpark@snu.ac.kr

Recent remarkable advancements in drug discovery have inspired the development of synthetic strategies aiming for the incorporation of bioactive skeletal features into newly synthesized molecules, as exemplified by late-stage functionalization[1,2], diversity-oriented synthesis (DOS)[3,4], biology-oriented synthesis (BIOS)[5], and bioconjugation reactions[6,7]. The key skeletal features of natural and synthetic bioactive compounds have been recognized as privileged structures with high biological relevance[8,9] and the incorporation of such structures into molecular frameworks has been an important criterion for harnessing their biological activities[10,11]. Therefore, the development of synthetic pathways for the incorporation of privileged structures into bioactive molecules, especially at the later stages of synthesis, may reveal bioactivities by expanding the chemical space toward unexplored biological applications via skeletal functionalization. For example, Wang et al. recently demonstrated the enhanced pharmacological effects (synergistic effects) of the products obtained by the incorporation of the skeletal features of clopidogrel (antiplatelet drug) into SC-560 (COX-1 inhibitor) via triazene-based cycloaddition, which showed that the conjugation of key skeletal features of drug moieties might induce synergistic effects[12]. On the other hand, despite the existing limitations[13,14], much effort has been directed at the synthesis of antibody-drug conjugates through the covalent conjugation of small molecules to biomacromolecules. However, the late-stage covalent conjugation of therapeutic agents to other bioactive small molecules, such as vitamins, amino acids, and sugars, has not been well explored[6,7].

In-line with our continuous efforts to develop synthetic routes for privileged substructure-based DOS (pDOS) strategies and thereby increase molecular diversity[10,15–17], we aimed to establish a methodological concept for the late-stage modification of various small molecules with biologically-relevant substructures, such as vitamins. In particular, we focused on vitamin $B_3$, also known as niacin or nicotinic acid, because of its simple structure and high biological relevance[18–21]. Nicotinates and nicotinamide are pharmaceutical surrogates of nicotinic acid[22], and are used as dietary supplements to treat pellagra and other vitamin $B_3$ deficiencies[20–22] in addition to being the key ingredients of numerous cosmetics and rubefacient creams[22–25]. Other classes of nicotinate drugs include vasodilators, estrogen, androgen (Testosterone nicotinate), opioid analgesic and cough suppressants (Nicocodeine), anti-inflammatory agents (Morniflumate), and hypolipidemic agents (Fig. 1a)[18,19,22]. On the other hand, (hetero) arylamines are key structural units that are found in numerous pharmaceuticals and feature a wide bioactivity range, as exemplified by their antipyretic, analgesic, anesthetic, antiviral, antibacterial, and anticancer activities (Fig. 1b)[2,26,27]. To the best of our knowledge, coupling reactions for connecting nicotinates to arylamines have not yet been reported. Although the Suzuki–Miyaura cross-coupling reaction is a standard protocol for constructing $C(sp^2)$–$C(sp^2)$ bonds from halogenated arylamines and appropriate arylboronic acids, this reaction has yet to be successfully applied to simple and/or electron-deficient pyridine boronic acids, which includes nicotinate or nicotinamide scaffold[28–30].

Because of the importance of the nicotinate moiety, numerous methods have been developed for its synthesis. Since the pioneering synthesis of 2-methylnicotinate by Dornow and Baumgarten in 1939 using β-alkoxyacrolein acetal and β-aminocrotonate[31], the Bohlmann–Rahtz synthesis[32], which employs substituted enamines and propargyl aldehydes/ketones, has been the major synthetic route to substituted pyridines, and its mechanistic principle has underpinned all further developments in the synthesis of nicotinates and other substituted pyridines, including the innovative Bagley modification of in situ α,β-substituted enamine generation (Fig. 1c)[33,34].

In this sequence, there are also several MCR (multicomponent reaction) approaches developed for synthesis of polysubstituted pyridines sharing nicotinate/nicotinamide scaffold[33,35–38]. However, the Bohlmann–Rahtz pyridine synthesis and its variants employ dicarbonyl compounds as enamine precursors and propargyl ketones or α,β-unsaturated carbonyls as the Michael acceptors, and therefore inevitably produce ortho-substituted nicotinates. To overcome this limitation, we carried out a careful analysis of the vitamin $B_3$ biosynthetic process, which inspired us to design a strategy involving the conjugation of nicotinates with (hetero)arylamines from simple (aza)indoles (Fig. 1d). This unparalleled biosynthetic process involves an enzymatic $C_2$=$C_3$ bond cleavage of the tryptophan indole ring to produce N-formyl kynurenine, an important structural intermediate that can be subsequently transformed into a nicotinic acid, which is also a precursor to other forms of vitamin $B_3$[20,21]. To simulate the formation of N-formyl kynurenine containing the aniline moiety released from the indole unit of tryptophan in the vitamin $B_3$ biosynthetic process, we hypothesized the skeletal remodeling of 3-formyl(aza)indole which allows the robust synthesis of meta-aminoaryl nicotinates with unfunctionalized $C_2$ and $C_6$ positions via aldol-type addition/dehydration and intramolecular cyclization followed by simultaneous re-aromatization and C–N bond cleavage (Fig. 1e and Supplementary Fig. 1).

Actually, highly regioselective direct C–H arylation at the $C_3$-position of substituted pyridines are possible (Fig. 1f)[39,40], only when pyridines contain electron-withdrawing groups (–NO$_2$, –CN, –F, and –Cl) or directing group (carboximide). In addition, these methods cannot be employed to conjugate functional aromatics like arylamines to pyridines. Furthermore, a retrosynthetic analysis of the cross-coupling-based routes to ortho-nicotinated arylamines also revealed serious limitations, including long synthetic routes, limited substrate scope due to the poor accessibility of corresponding boronic esters, and apparent competitive side reactions (Fig. 1g). In fact, halogen (Br, I)- as well as boronic ester-substituted products, which could be valuable substrates for further diversification, proved impossible to access, thereby placing serious constraints on transition metal-based cross-coupling strategies. This proposed biomimetic strategy allowed the transformation of (aza)indole scaffolds to the meta-amino(hetero)aryl nicotinates in good yields and featured a broad substrate scope with limited side reactions while offering unique access to halogen/boronic ester-substituted (hetero)arylamines conjugated to vitamin $B_3$ along with robust conjugation of arylamines and nicotinate pharmaceuticals (Fig. 1h). This unique transformation involves the sequential reactions of different intermediates generated in the one-pot strategy and almost exclusively yields a single major product at the end of the final step. Although a number of research groups have attempted indole ring transformations to aminoaryl pyrazoles or other fused ring scaffolds[17,41–43], in this work, we show the robust biomimetic remodeling of (aza)indoles to the vitamin $B_3$ scaffolds conjugated to (hetero)arylamines as the last-stage transformation.

## Results

**Reaction optimization and substrate scope.** We initially investigated our hypothesis (shown in Fig. 1e) for the remodeling of 3-formylindoles with the readily available β-aminoacrylate; however, the reaction did not proceed efficiently (<10% conversion, see Supplementary Figs. 1 and 3). Therefore, we postulated the in situ generation of β-aminoacrylate from the corresponding propiolates with NH$_4$OAc, which is known to form cationic complexes by activating the alkyne triple bond via cation-π interactions[44] and also act as an ammonium source for the enamine synthesis[33,34] (Supplementary Fig. 4). After a careful investigation with initial trials, we selected N-phenylsulfonyl-7-azaindole-3-carboxaldehyde (**1a**) and ethyl propiolate (**2a**) as

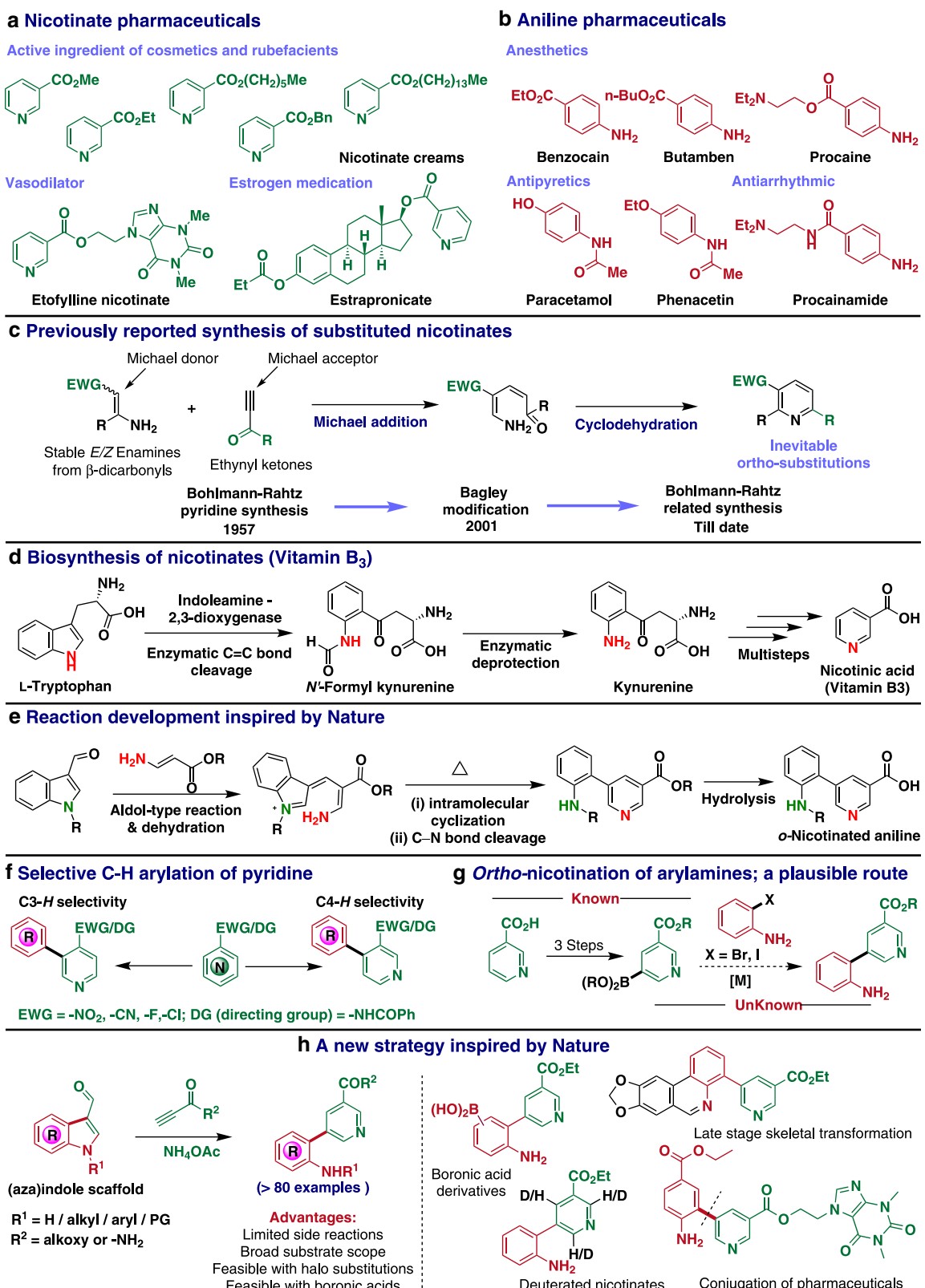

**Fig. 1 Background for reaction development. a** Representative examples of commercially available bioactive nicotinates. **b** Representative examples of aniline-based pharmaceuticals. **c** Previously reported synthetic routes for substituted nicotinates with inevitable $C_2$ and $C_6$ substitutions. **d** Biosynthetic process of vitamin $B_3$ (a.k.a niacin or nicotinic acid) from L-tryptophan. **e** Reaction development with working hypothesis, which mimics the nature's pathway for *ortho*-nicotinated anilines as the late-stage modification with vitamin $B_3$ without substituents at the $C_2$ and $C_6$ positions. **f** Regioselective direct C–H arylation of pyridines containing electron-withdrawing or directing groups. **g** Plausible synthetic route for the conjugating anilinic compounds with vitamin $B_3$ on the basis of the literature evidences. **h** This study: synthetic strategy for conjugating anilinic compounds with vitamin $B_3$ by mimicking nature's biosynthetic pathway and its advantages.

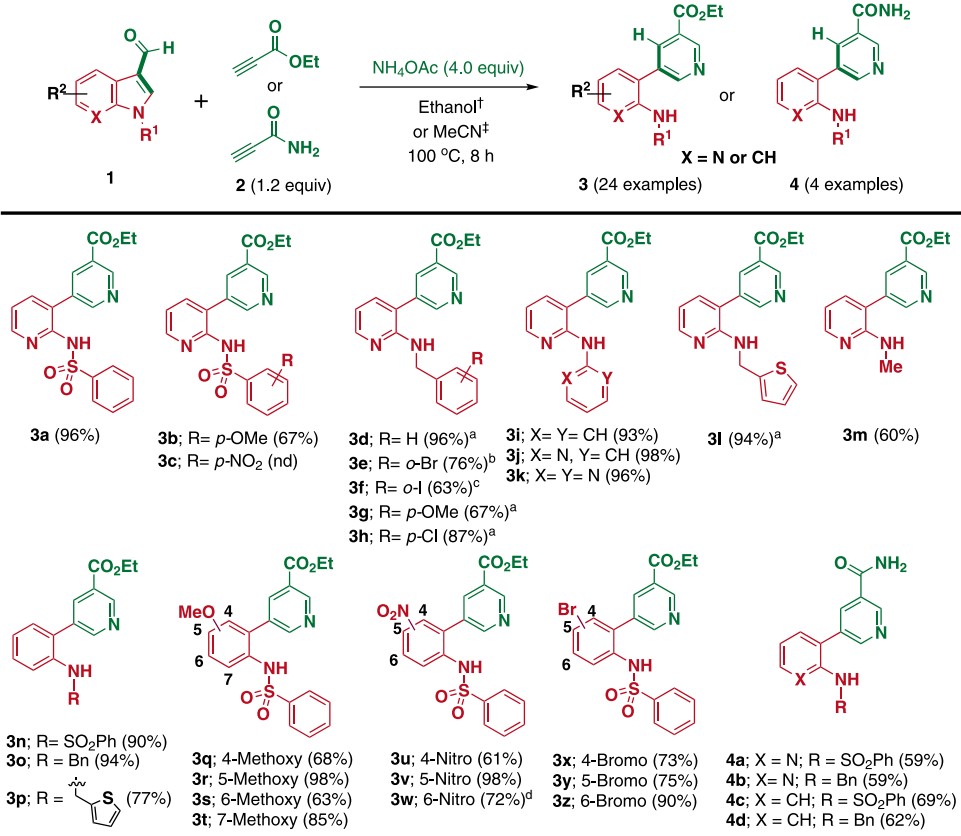

**Fig. 2 Substrate scope for *N*-substituted (aza)indole remodeling.** Reaction conditions: **1** (0.2 mmol), **2** (1.2 equiv.), $NH_4OAc$ (4.0 equiv.) in an appropriate solvent (2 mL) at 100 °C for 8 h. Yields of isolated products (**3** and **4**) are given in parenthesis. nd = not detected. [†]Ethanol was used as the solvent for all of the reactions except for starting materials containing sulfonyl-protected indoles. [‡]Acetonitrile was used as the solvent in the case of sulfonyl-protected indoles as substrates. [a]Reaction were performed at 0.5 mmol scale. [b]2.0 g scale reaction with respect to **1**. [c]500 mg scale reaction with respect to **1**. [d]Reaction time 16 h.

the benchmark substrates for reaction optimization, wherein $NH_4OAc$ was used as the ammonium source for in situ enamine generation. Following the investigation of various parameters (Supplementary Table 1), we determined the optimal reagent quantities to be 1.0 M **1a** in EtOH containing **2a** (1.2 equiv.) and $NH_4OAc$ (4.0 equiv.). The reaction conditions involved heating the mixture at 100 °C, which yielded the desired product **3a** (Fig. 2) in an average yield of 96%. Under the optimized conditions, we first explored the reactivity of the diversely *N*-substituted 7-azaindole-3-carboxaldehydes (**1a–1m**) containing sulfonamide, benzyl, aryl, and alkyl groups. Various *N*-substituted 7-azaindole-3-carboxaldehydes formed the desired *meta*-2-aminopyridyl nicotinates (**3a–3m**) in good to excellent yields, indicating that the electronic nature of the substituents at the azaindole $N_1$ position did not significantly affect the product yields or the ring transformation kinetics under the optimized conditions. However, *N*-acyl 7-azaindole-3-carboxaldehydes showed the significant deacylation under the optimized condition (Supplementary Fig. 2).

After establishing the azaindole-based substrate scopes, we turned our attention to the indole derivatives. During the investigation of the substrate scope with *N*-sulfonylated indole derivatives as starting materials, we observed considerable byproduct formation, to address which, we performed additional solvent optimization (Supplementary Table 2). In the case of the *N*-sulfonylated indole derivatives, $CH_3CN$ was identified as the best solvent for the reaction in the sealed vial at 100 °C, which delivered the products with minimal byproduct formation, despite ethanol being optimal for all other *N*-substituted (aza)

indoles. With this condition, *N*-phenylsulfonyl, *N*-benzyl, and *N*-thiophen-2-ylmethyl indole-3-carboxaldehydes (**1n–1p**) were smoothly transformed into the desired products **3n**, **3o**, and **3p** in yields of 90, 94, and 77%, respectively.

Unlike azaindoles, numerous substituted indoles are commercially available, thereby allowing for a broader substrate scope study. Therefore, the reaction compatibility with diverse electronic natures of the substituents on the indole rings was explored by placing methoxy, nitro, and bromo groups at various positions in the structure of **1n**. Interestingly, all of the substrates (**1q–1z**) transformed to the corresponding desired products (**3p–3z**) in good to excellent yields irrespective of the electronic nature of the substituents, as well as the site of substitution. Notably, the formation of **3w** required a longer reaction time (16 h) to reach the completion. However, the $C_7$ nitro- and bromo-substituted products were not obtained since the sulfonyl protection of the corresponding indole-3-carboxaldehydes was not successful. Furthermore, we expanded the reaction scope to the synthesis of *meta*-substituted nicotinamides **4a–4d** in moderate to good yields by replacing ethyl propiolate with propiolamide as a reaction partner.

**Use of (aza)indole-3-carboxaldehydes with a free –NH group.** Encouraged by the versatility of the cascade ring transformation, we explored the possibility of ring cleavage of 3-formyl(aza) indoles bearing a free –NH group. To our delight, under the above optimized conditions, **6a** and **6f** were produced from 7-azaindole-3-carboxaldehyde (**5a**) and indole-3-carboxaldehyde

(5 f) in 60 and 66% [1]H NMR yields, respectively. However, the yields were not consistent over multiple runs, and could not be improved by increasing the reaction time. Subsequently, we modified the reaction conditions to activate the free –NH and formyl groups by the addition of suitable Lewis acid catalysts. The catalyst screening revealed that the addition of 10 mol% Zn(OTf)$_2$ at a slightly elevated temperature of 120 °C significantly improved the isolated yields of 6a (90% avg) and 6f (90% avg) in ethanol and CH$_3$CN, respectively, with excellent consistency (Supplementary Table 3 and 4). Indeed, the forte of our biomimetic strategy allows the use of a wide range of readily available (aza)indole-3-carboxaldehydes with free –NH group to directly access *ortho*-nicotinated free anilines or aminopyridines. The smooth transformation of all possible types of azaindole-3-carboxaldehydes (5a, 5c–5e) to the corresponding products (6a, 6c–6e) provided significant diversity in the resulting heterobiaryl skeletons. Moreover, the easily accessible 4-chloro-7-azaindole-3-carboxaldehyde (5b) was employed to synthesize 6b which could be further diversified.

Indeed, the availability of a wide variety of indole derivatives allows the diversification of the corresponding *meta*-aminoaryl nicotinates in all possible substitution patterns around the indole scaffold. We then explored the substrate scope by extensively varying the substituents; methoxy (5g–5j), nitro (5k–5n), fluoro (5o–5r), bromo (5s–5 v), and pinacol boronate (5z–5ac) were evaluated at each of the C$_4$–C$_7$ positions of the indole-3-carboxaldehydes. Further, by considering the important roles of CF$_3$ moiety in drug discovery, we also evaluated the substrate scope with CF$_3$-substituted indole-3-carboxaldehydes (5w–5y). As shown in Fig. 3, all substrates were transformed into the corresponding desired products (6g–6ac) in moderate to good yields. Similarly, under the optimized conditions, 5a and 5f readily reacted with propiolamide to afford 2-aminopyridyl (7a) and 2-aminophenyl nicotinamides (7b), respectively. It is worth mentioning that the synthesis of the bromo- and boronic

ester-substituted products (3x–3z, 6s–6v, and 6z–6ac) makes our biomimetic strategy more attractive and showcases its versatility. These bromo- and boronic ester-derived products cannot be accessed by any other cross-coupling strategies, and are important substrates for further diversification at all possible positions around the aniline scaffold and thereby allow significantly enhanced exploration of the unexplored chemical space.

As shown in Figs. 2 and 3, we observed no considerable differences in the reactivity among substrates containing various substituents under the optimized conditions, which revealed that the reaction mechanism might feature more than one rate-limiting step in the sequential reaction pathway, and the overall reaction rate across the different steps may converge to a similar value irrespective of substrate substitution pattern. Therefore, a series of experiments were performed in parallel under identical reaction conditions featuring a short reaction time of 30 min to examine the initial rate of product formation, which may shed light on the influence of the electronic effects on the substrate reactivity. As shown in Supplementary Table 5, azaindole-3-carboxaldehyde (5a) was more reactive than the corresponding indole-3-carboxaldehyde (5f), and the nitro-substituted substrates (5l, 5m) were more reactive than the corresponding methoxy-substituted ones (5h, 5i). Therefore, the presence of electron-withdrawing groups on the indole ring increases their reactivity, possibly due to the increased rate of the C$_2$–N bond cleavage.

**Synthetic application**. The developed nature-inspired synthetic strategy is based on the idea of conjugating bioactive small molecules with other privileged structures in the later stages of the synthesis, and this biomimetic route was used to establish a synthetic method for forging bioactive (hetero)arylamines conjugated with vitamin B$_3$ (Fig. 4). For demonstration purposes, our biomimetic strategy was employed to construct a handful of covalently fused bioactive entities from the 3-formyl(aza)indole derivatives and

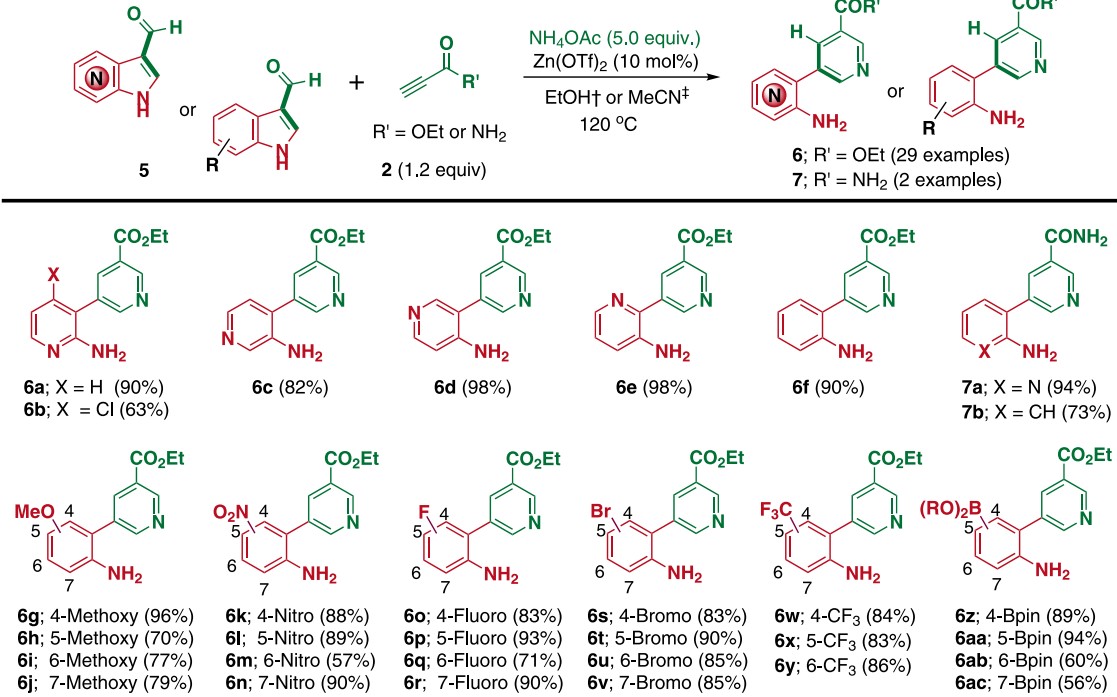

**Fig. 3 Substrate scopes for synthesis of nicotinated anilines/aminopyridines via (aza)indoles skeleton remodeling.** Reaction conditions: 5 (0.2 mmol), 2 (1.2 equiv.), NH$_4$OAc (5.0 equiv.), and Zn(OTf)$_2$ (10 mol%) at 120 °C. †Reactions were performed in EtOH (2.0 mL) for 6 h in the case of azaindole-based substrates (6a–6e, 7a). ‡Reactions were performed in CH$_3$CN (2.0 mL) for 16 h in the case of indole-based substrates (6 f–6ac, 7b). Yields of the isolated products (6 and 7) are reported.

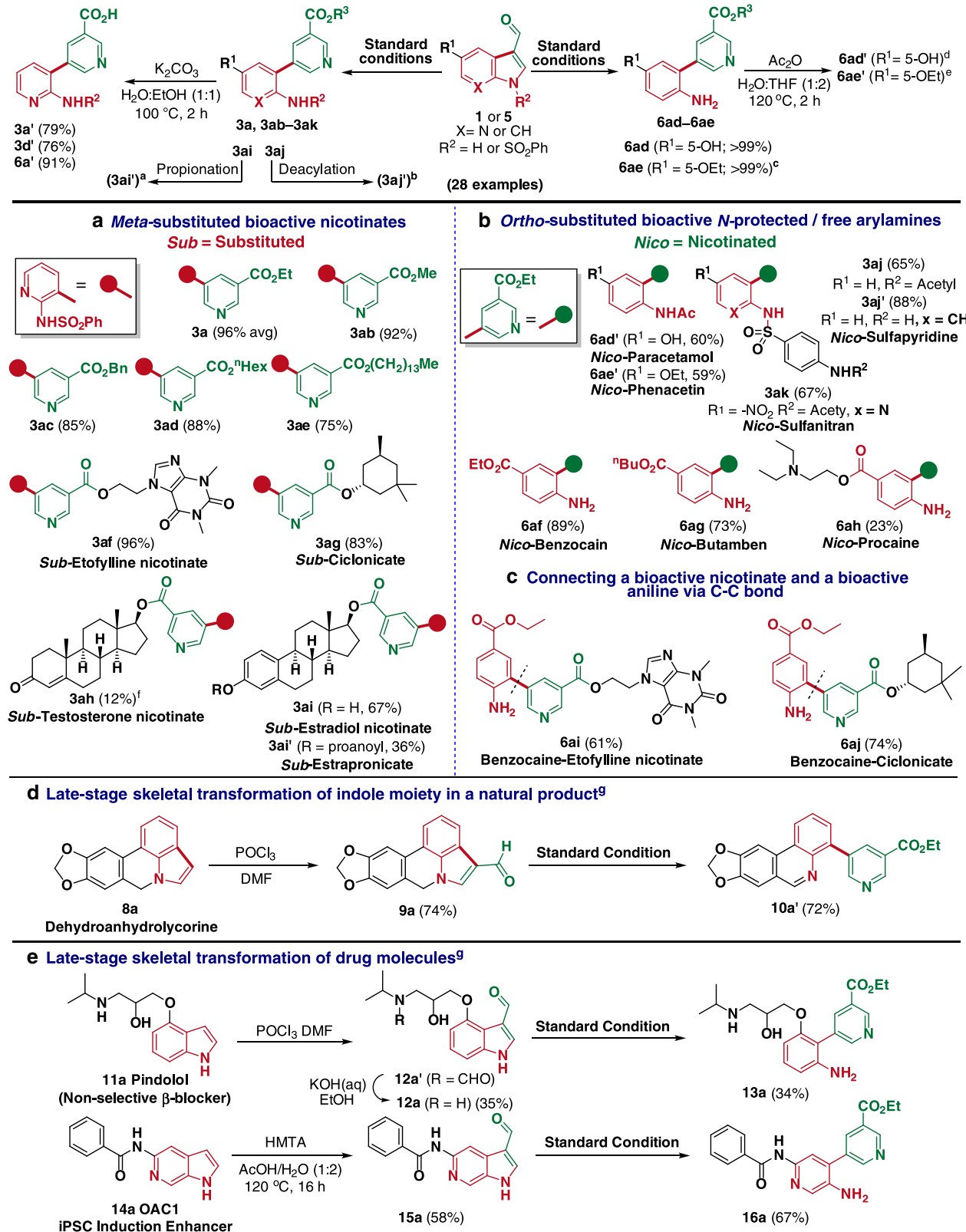

**Fig. 4 Extending the scope of the reaction towards the synthesis of substituted bioactive niacinates and anilinic drugs. a** Scope of the reaction for the synthesis of *meta*-substituted bioactive nicotinates. **b** Scope of the reaction for the synthesis of *ortho*-substituted bioactive *N*-protected or free anilinic drugs. **c** A representative example of conjugates containing bioactive nicotinates and anilinic drugs via C–C bond. **d**, **e** Extension of the scope of our synthetic strategy for late-stage skeletal transformation of (aza)indole moiety in natural products and pharmaceuticals. See Supplementary Information for reaction condition and further details[(a–g)].

the appropriate propiolates. Diverse pharmaceutically-active nicotinates were successfully linked to *N*-phenylsulfonyl-2-aminopyridines at its *meta* position. The *N*-phenylsulfonyl-protected 2-aminopyridine scaffold (red dot in Fig. 4a) was selected in preference to simple anilines or other aminopyridines, owing to its presence in many sulfa-based drugs, including sulfasalazine, which is designated as an essential medicine by the World Health Organization (WHO). Under the optimized conditions, **1a** reacted with the appropriate propiolates (**2a**–**2i**) to afford products **3a** and **3ab**–**3ae** (75–96%), in which the *meta* position of the pharmaceutically-active nicotinates (ethyl, methyl, benzyl, *n*-hexyl, and myristyl nicotinates), known to be key ingredients in cosmetics and rubefacient creams, were covalently conjugated with *N*-phenylsulfonyl 2-aminopyridine scaffolds. In a similar fashion, we obtained **3af** (96%) and **3ag** (83%) in which the *meta* position of the vasodilating agents, such as etofylline nicotinate and ciclonicate, were conjugated with the *N*-phenylsulfonyl 2-aminopyridine scaffold. Furthermore, testosterone propiolate and estradiol propiolate reacted with **1a** to afford **3ah** (12%) and **3ai** (67%), bearing scaffolds of testosterone nicotinate (an androgenic steroid) and estradiol nicotinate, respectively. **3ai** was further propionated to afford **3ai′** bearing the estrapronicate (an estrogenic steroid) scaffold.

The presented biomimetic strategy also provides robust access to various amine pharmaceuticals that are covalently fused with ethyl nicotinate (green dot in Fig. 4b). Under the optimized conditions, ethyl propiolate was coupled with hydroxy- and ethoxy-substituted indole-3-carboxaldehydes to afford **6ad** and **6ae**, which were acetylated without further purification to furnish paracetamol **6ad′** (60%) and phenacetin **6ae′** (59%), in which the ethyl nicotinate is covalently conjugated at the *ortho* position. Similarly, *N*-acetanilide-*p*-sulfonyl-7-azaindole-3-carboxaldehydes **1aj** and **1ak** were transformed to **3aj** and nicotinated sulfanitran **3ak**, respectively, while nicotinated sulfapyridine **3aj'** was obtained by the deacylation of **3aj**. Indole-3-carboxaldehydes substituted with various esters and amides at the $C_5$ position afforded **6af**, **6ag**, and **6ah**, which feature the key scaffolds of local anesthetics, namely benzocaine, butamben, and procain, respectively. As vasodilation can be associated with a local anesthetic response[45,46], the conjugation of vasodilators and local anesthetics is expected to result in a synergistic effect. The developed synthetic method was therefore employed for conjugating benzocaine to etofylline nicotinate and ciclonicate at the *meta* position to afford **6ai** and **6aj**, respectively (Fig. 4c). Further, to explore the feasibility of product diversification, we further transformed the nicotinate scaffold in **3a**, **3d**, and **6a** into vitamin $B_3$ scaffold by hydrolysis, which afforded **3a′**, **3d′**, and **6a′** in good to excellent yields of 79, 76, and 91%, respectively.

**Late-stage remodeling of (aza)indole core.** Next, we attempted the late-stage skeletal transformation strategy to the indole-based natural products (Fig. 4d) and therapeutic agents (Fig. 4e). For example, naturally occurring dehydroanhydrolycorine **8a**[47] and therapeutic agents such as pindolol **11a**[48] (non-selective *β*-blocker) and OAC1 **14a**[49] (iPSC induction enhancer) were formylated to afford **9a**, **12a**, and **15a**, respectively. These compounds successfully underwent a skeletal remodeling of the (aza)indoles under the standard reaction conditions. Interestingly, the expected product **10a** from **9a** was found to be susceptible to spontaneous oxidation at the benzylic position, and the analysis of the crude reaction mixture showed the formation of both **10a** and its oxidized product **10a′**. However, **10a** underwent spontaneous oxidation during silica-gel column purification to exclusively yield the oxidized product **10a′** (72%). In the case of **11a**, we obtained the inevitable *N*-formylated intermediate **12a′** via Vilsmeier-Haack reaction, and hydrolytic deformylation allowed the conversion of **12a′** to **12a**. Our skeletal transformation strategy allowed the formation of unique heterobiaryl product **13a** (34%) via late-stage remodeling of **11a**. Similarly, the formylated iPSC induction enhancer **15a** underwent a ring transformation under the standard reaction conditions and afforded the heterobiaryl product **16a** in good yield (67%).

**Mechanistic studies.** Based on our working hypothesis and literature precedence, we proposed two different plausible reaction pathways (Fig. 5a) for the skeletal transformation of 3-formyl (aza)indoles to *meta*-aminoaryl nicotinates. To reveal the favorable mechanistic pathway, we designed a series of deuterium-labeling experiments (Fig. 5b). Based on our NMR analyses ([1]H, [13]C, COSY, and HSQC) and literature reports, we assigned the [1]H chemical shifts of the nicotinate scaffold of **6f**. When we performed this transformation with the individually deuterated substrates **5f′** and **5f″**, and clearly observed the disappearance of [1]H peaks at δ 8.4 ppm in the product **6f′** and those at δ 8.8 ppm in **6f″**, respectively. These results revealed that our biomimetic (aza)indole transformation follows the reaction pathway B, initiated by the Aldol-type addition (Supplementary Fig. 5). Interestingly, as stated earlier, the commercially available *β*-aminoacrylate did not react efficiently with 3-formyl(aza)indoles, but the addition of $NH_4OAc$ slightly improved the [1]H NMR yield to 35% (Supplementary Fig. 6). Therefore, $NH_4OAc$ may also play an important role in driving this reaction forward by serving as a catalyst for the Aldol-type addition. Finally, we performed our skeletal transformation of **5f** under standard conditions using predeuterated ethyl propiolate in $C_2H_5OD$ and confirmed the formation of the deuterated product **6f‴** by [1]H NMR, which additionally supports our reaction mechanism as pathway B.

In conclusion, a unique biomimetic strategy was developed to conjugate vitamin $B_3$ with pharmaceutically important (hetero) arylamines, and the significance of this methodology lies in the late-stage conjugation of various aniline-based pharmaceuticals to nicotinates and vice versa. Owing to the wide occurrence of (aza) indole scaffolds in natural products and pharmaceuticals, the presented approach provides an excellent opportunity for remodeling the skeletal features of (aza)indole-containing pharmaceuticals and natural products, and also for constructing new molecular frameworks to expand the unexplored chemical space. Most importantly, the above strategy disclosed a mechanistic pathway different from that of the Bohlmann–Rahtz pyridine synthesis, which can provide insights to explore new reactions in pyridine synthesis and overcome the limitations of previous synthetic methods developed in the last six decades. Further biological evaluations of the resulting vitamin $B_3$-(hetero) arylamine conjugates are currently ongoing in our laboratories, and the results will be reported in due course.

## Methods

**General procedure for the reaction of N-substituted (aza)indole-3-carbox-aldehydes.** A 4-mL vial equipped with a magnetic stir bar and a Teflon-lined screwed cap was charged with **1** (0.2 mmol), ethyl propiolate (1.2 equiv.), and $NH_4OAc$ (61.66 mg, 4 equiv.) in the appropriate solvent (2.0 mL, EtOH or $CH_3CN$). The vial was then sealed and heated at 100 °C for 8 h. Upon reaction completion checked by TLC analysis, the reaction mixture was concentrated under the reduced pressure, added with saturated aqueous $NaHCO_3$, and extracted with dichloromethane (DCM, 3 × 10 mL). The combined organic fractions were dried over anhydrous $Na_2SO_4$, filtered, and concentrated under reduced pressure. The crude compound was purified by silica-gel flash column chromatography to obtain the desired product bearing the nicotinate (**3**) or the nicotinamide (**4**) scaffolds.

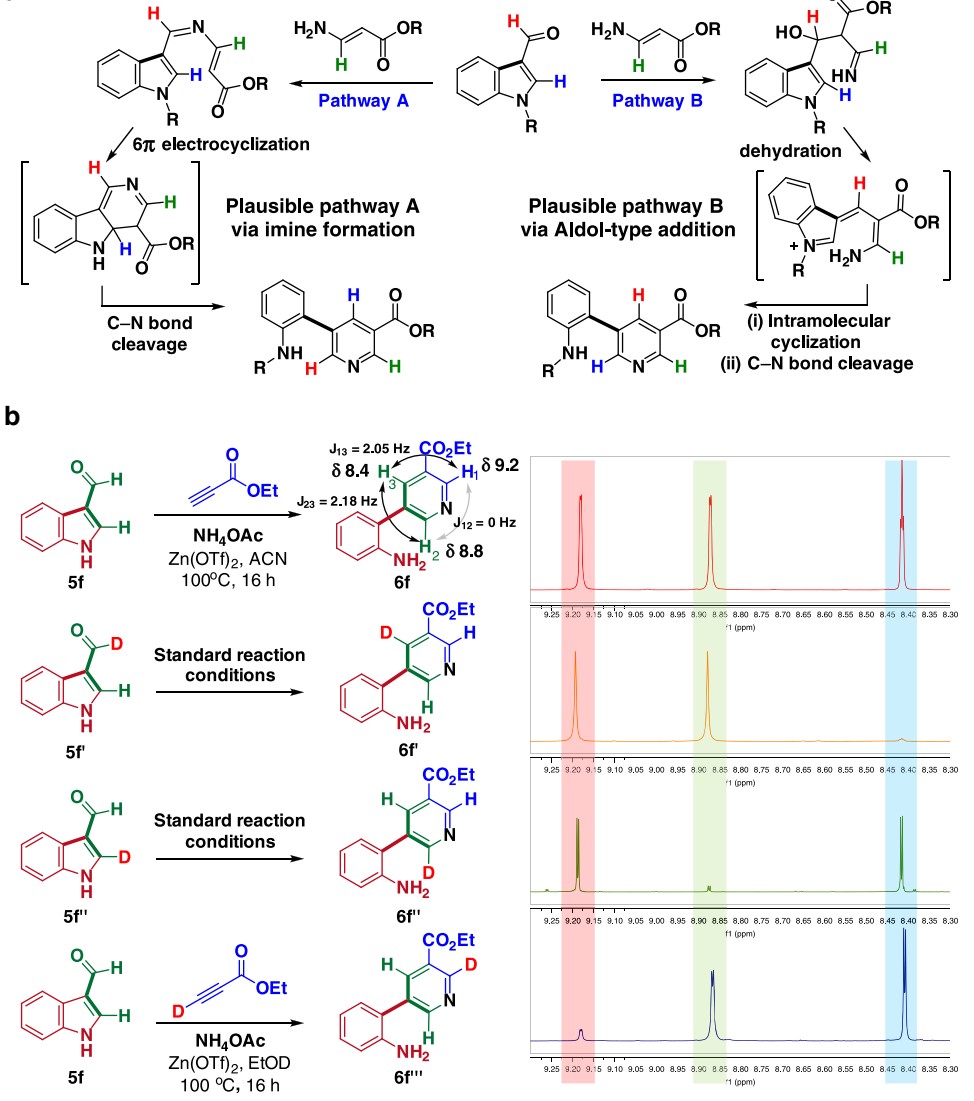

**Fig. 5 Plausible mechanisms of nature-inspired transformation of (aza)indole scaffolds into substituted *meta*-aminoaryl nicotinate scaffolds.**
**a** Plausible mechanism A via imine formation and 6π electrocyclization followed by C–N bond cleavage and re-aromatization. Plausible mechanism B via Aldol-type addition, dehydration, and intramolecular cyclization followed by C–N bond cleavage and re-aromatization. Each proton in *meta*-aminoaryl nicotinates was color-coded to track its sources in starting materials. **b** Deuterium-labeling experiments revealed that our structural remodeling of (aza) indoles into substituted *meta*-aminoaryl nicotinate scaffolds occurs via plausible pathway B, as confirmed by NMR experiments.

**General procedure for the reaction of (aza)indole-3-carboxaldehydes**. A 4-mL vial equipped with a magnetic stir bar and a Teflon-lined screwed cap was charged with **5** (0.2 mmol), **2** (1.2 equiv.), NH$_4$OAc (77.08 mg, 5 equiv.), and Zn(OTf)$_2$ (7.27 mg, 0.1 equiv.) in the appropriate solvent (2.0 mL, EtOH or CH$_3$CN). The vial was then sealed and heated at 120 °C for the desired times (6–16 h). Upon reaction completion checked by TLC analysis, the reaction mixture was concentrated under the reduced pressure, added with saturated aqueous NaHCO$_3$, and extracted with DCM (3 × 10 mL). The combined organic fractions were dried over anhydrous Na$_2$SO$_4$, filtered, and concentrated under reduced pressure. The crude compound was purified by silica-gel flash column chromatography to obtain the desired products bearing the nicotinate (**6**) or the nicotinamide (**7**) scaffolds.

## Data availability
All data generated and analysed during this study are included in this article and its Supplementary Information, and also available from the authors upon reasonable request.

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

## Acknowledgements

This work was supported by the Creative Research Initiative Grant (2014R1A3A2030423) and the Bio & Medical Technology Development Program (2012M3A9C4048780) through the National Research Foundation of Korea (NRF) funded by the Korean Government (Ministry of Science & ICT). S.Y. is grateful for the predoctoral fellowship by NRF-Fostering Core Leaders of the Future Basic Science Program/Global Ph.D. Fellowship Program (2017H1A2A1045200).

## Author contributions

B.V.V. and S.B.P. conceived, designed, and originated this project. B.V.V., K.V., and S.Y. performed the experiments, obtained all spectroscopic data, and analyzed the results. B.V.V., K.V., and S.B.P. co-wrote the manuscript. All authors analyzed the data, discussed the results and contributed to the relevant discussion.

## Competing interests

The authors declare no competing interests.
