## [Peer Review File · Nature Communications]

REVIEWER COMMENTS

Reviewer #1 (Remarks to the Author):

The manuscript describes interesting ring-opening of azaindole-3 carboxaldehyde. I agree with authors that nicotinic acid derivatives are important compounds. I therefore strongly believe that this approach will be useful to access ortho-unsubstituted nicotinates which can be used for synthesis of other biologically active compounds. The burden of the contribution should therefore lie in the novelty and importance of the mechanism. Also in that regard, I am not so convinced by the paper.

The should clarify the following doubts-

1. The ring-opening of indoles is interesting but not surprising. I would argue that the starting materials used, that are sulfonyl azaindoles and Acetyl azaindoles are well documented. Most of the starting materials are very similar to those reported in ACS Comb. Sci. 2018, 20, 10, 573–578. Manna and co-workers used indole, here Park and co-authors used azaindole. Sulfonyl indole and Acetyl indoles are regarded as very reactive olefins, which naturally undergo various addition reactions. The initial concept of using sulfonyl or acyl protection was already reported in ACS Comb. Sci. 2018, 20, 10, 573–578. So, the authors should justify the novelty of their approach.

2. The authors also used Lewis acid like Zn(OTf)₂ instead of BF₃.OEt₂ because the difference in product yield was 3% only (which is within the error limit). BF₃.OEt₂ is much cheaper than Zn(OTf)₂, so the authors should justify why they have selected Zn(OTf)₂ over BF₃.OEt₂.

3. The authors should highlight the chemistry reported in ACS Comb. Sci. 2018, 20, 10, 573–578. The synthesis of ortho-unsubstituted nicotinates from azaindole is an extension of the previous works.

4. Authors suggest that C2-N bond cleavage is the driving force of the reaction. However, there is no significant proof for their claim. The authors should provide the proof for the formation of the intermediates. Only ¹H NMR based studies not enough for their claim. The provided ¹H NMR experiment does not completely support the given mechanism. We found the mechanism is not fully supporting the way authors have proposed.

5. The in-situ generation of β-aminoacrylate from the corresponding alkyne in the presence of NH₄OAc is really a new concept?

6. If this is really a novel approach, then why the authors have not mentioned about other nicotine derivatives except nicotinates only.

7. Reference to the opening of the C-N bond of indoles via the addition of two nucleophiles at the C2 positions should be mentioned such as J. Org. Chem. 1960, 486.

After carefully addressing the doubts, the paper should be resubmitted to Nature Communications.

Reviewer #2 (Remarks to the Author):

In this manuscript, Park and coworkers report development of a new strategy for the synthesis of meta-aminoaryl nicotinates from 2 formyl(aza)indoles. Unlike previous synthetic routes, this biomimetic method utilizes propiolates as enamine precursors and thus allows access to ortho-unsubstituted nicotinates. Additionally, this method enables the late-stage installation of bioactive (hetero)arylamines with nicotinates and nicotinamides. Most importantly, this process allows to

produce a wide range of bisheterocyclic compounds that are difficult to access by the use of known methods, which is very useful for medchem studies and pharmaceutical industry. The authors also perform mechanistic studies in order to propose the mechanism for product formation. It's a good piece of work and I recommend its publication in Nature Communication after the following minor concerns are addressed.

- 1) The authors should add one scheme to show the methods of direct functionalization of C3 of pyridine, and briefly give a comparison with your method. (J. Am. Chem. Soc. 2011, 133, 6964; J. Am. Chem. Soc. 2011, 133, 16338)
- 2) The authors should add 2-4 examples of useful substituents such as -F, -CF₃.
- 3) The authors should add additional results of 3-formylbenzofuran as the starting material.
- 4) Is the ester or amide on the alkyne 2 necessary for the reaction to take place? Author should add the results of the other electron-withdrawing group on the alkyne, for example: -Bpin, -CN.
- 5) In the supporting information, the authors should update the spectrum of product 5b-13C, 9a-13C, 6i-13C, 6d-13C, 6h-13C, 6j-13C/1H, 10a-13C.

Reviewer #3 (Remarks to the Author):

The manuscript submitted by Park and coworkers constitutes an elegant and very complete piece of work in organic synthesis, allowing access to original molecules combining two motifs whose biological activity is recognized, a nicotinate motif and an aniline motif.

The developed method consists of a three-component reaction between a 3-formylindole or azaindole, a beta-aminoacrylate and ammonium acetate, resulting in a domino sequence aldol-type addition / intramolecular cyclization / CN bond cleavage / re-aromatization. The products thus obtained are nicotinate or nicotinamide derivatives having a (hetero) arylamine substituent in the meta position. The substrates and reagents are simple and easily available, the reaction conditions are easy to implement, and the products are obtained with good to very good yields. The procedures and product characterizations are described in great detail in the supporting information, so that the experiments should be reproducible without difficulty.

Even if at first glance the sequence could be considered as a variant of the Bohlmann-Rahtz reaction, the originality lies in the use of indoles (or azaindoles) which rearrange themselves via the breaking of a C-N bond, releasing on the one hand the nicotinate or nicotinamide nucleus, and on the other hand the primary amine function of the aniline unit. This rearrangement is based on a clever working hypothesis which is inspired by the biosynthesis of vitamin B3 from L-Tryptophan. The reaction mechanism has been clearly demonstrated by NMR studies, notably using deuterated substrates.

The only downside that we could bring to this manuscript, considering in particular the strong experience of this team in the field of the development of new compounds for well-targeted biological applications, is the lack of discussion and arguments aimed at demonstrate the value of combining the two bioactive motifs concerned, for example by identifying a biological barrier for which these new products could provide solutions. Nevertheless, due to the nature of the two bioactive motifs thus conjugated, which revealed numerous and diverse biological activities, this work can reveal new bioactive compounds, which can be considered as derivatives of vitamin B3, by expanding the chemical space towards unexplored biological applications.

I support the publication of this article without any particular modification. Nevertheless, if we consider that the synthetic approach developed consists of a three-component reaction, other work leading to polysubstituted pyridines according to an MCR approach, including the formation of nicotinamide-type derivatives, could be cited in the references. We will retain for example the work of the Menendez team (ACS Comb. Sci. 2012, 14, 551) or those of the Rodriguez and Constantieux

team (Chem. Commun. 2008, 4207 ; Chem. Eur. J. 2009, 15, 12945 ; Adv. Synth. Catal. 2012, 354, 2537)

Point-by-Point Responses to Reviewers' Comments

We are very thankful to the reviewers' insightful comments and valuable inputs, which helped us to improve the manuscript. All reviewers were principally positive about our study and stated that our study was found to be interesting and educative to read. On the basis of the suggestions provided by three reviewers, we revised the manuscript and supporting information to sincerely address most of issues raised by all reviewers.

Reviewer #1

The manuscript describes interesting ring-opening of azaindole-3-carboxaldehyde. I agree with authors that nicotinic acid derivatives are important compounds. I therefore strongly believe that this approach will be useful to access ortho-unsubstituted nicotinates which can be used for synthesis of other biologically active compounds. The burden of the contribution should therefore lie in the novelty and importance of the mechanism. Also, in that regard, I am not so convinced by the paper.

Response:

We appreciate the reviewer #1 for his/her positive analysis by identifying the usefulness of our approach. For the novelty issue, we could attain more clarity by realizing the present work as a unique biomimetic approach to synthesize a challenging meta-aminoaryl nicotinate scaffolds, along with perceiving the useful access to ortho-unsubstituted nicotinates. The same was recognized and appreciated by reviewer #2 and #3. Also, we discussed about this in more details in our justification letter.

In this regard we would like to quote the statements from reviewer #2 and reviewer #3

"...The manuscript submitted by Park and co-workers constitutes an elegant and very complete piece of work in organic synthesis, allowing access to original molecules combining two motifs whose biological activity is recognized, a nicotinate motif and an aniline motif ..."
(reviewer #3)

"...Most importantly, this process allows to produce a wide range of bisheterocyclic compounds that are difficult to access by the use of known methods, which is very useful for medchem studies and pharmaceutical industry...." (reviewer #2).

Therefore, the novelty of the present chemistry comes from the installation of nicotinates to the ortho-position of functionalized anilines which has never been reported till now. The present study is originated from the hypothesis on the basis of transformation of the indole unit of tryptophan in the vitamin B₃ biosynthetic process, which lead us to the development of robust chemistry aimed to expand the unexplored biorelevant chemical space of nicotinates conjugated with bioactive (hetero)aryl amines. We demonstrated the strength of our methodology by conjugating nicotinate motif to prominent pharmaceuticals (>20 examples) and showing late-stage skeletal transformations of natural product and pharmaceuticals. The easy accessibility toward halo- or boronic acid-derived bisheterocyclic products further expands the utilities of this biomimetic chemical transformation. Therefore, we firmly believe that this chemistry will surely harvest the greater impact in pharmaceutical industries, especially for library construction of drug-like bioactive heteroaryl conjugates with nicotinates.

Further, we would like to clarify the importance of our mechanism. Actually, reviewer #2 and #3 recognized the importance of our reaction mechanism by stating the following comments;

“... Most importantly, this process allows to produce a wide range of bisheterocyclic compounds that are difficult to access by the use of known methods, which is very useful for medchem studies and pharmaceutical industry. The authors also perform mechanistic studies in order to propose the mechanism for product formation... (reviewer #2)”

“... Even if at first glance the sequence could be considered as a variant of the Bohlmann-Rahtz reaction, the originality lies in the use of indoles (or azaindoles) which rearrange themselves via the breaking of a C-N bond, releasing on the one hand the nicotinate or nicotinamide nucleus, and on the other hand the primary amine function of the aniline unit. This rearrangement is based on a clever working hypothesis which is inspired by the biosynthesis of vitamin B3 from L-Tryptophan. The reaction mechanism has been clearly demonstrated by NMR studies, notably using deuterated substrates... (reviewer #3)”

The essential element associated with the mechanism is the plausible reaction pathways (Fig. 5a in the original manuscript). The key step in Bohlmann-Rahtz pyridine synthesis is the imine condensation to form a cyclized product, and generally a high reactivity is expected between amine and aldehyde. Interestingly, as described in Fig. 5a, pathway A and B are the most plausible mechanisms and both provide the same product. To elucidate the exact mechanism of this chemical transformation (Fig. 5b), we carefully designed the deuterated substrates that clearly showed that our transformation is initiated by Aldol-type condensation (pathway B). Hence, we believe that our mechanistic investigation provides the insight to explore new reactions in pyridine synthesis along with Vitamin B₃ analogues.

Comments from the reviewer #1

1. The ring-opening of indoles is interesting but not surprising. I would argue that the starting materials used, that are sulfonyl azaindoles and Acetyl azaindoles are well documented. Most of the starting materials are very similar to those reported in ACS Comb. Sci. 2018, 20, 10, 573–578. Manna and co-workers used indole, here Park and co-authors used azaindole. Sulfonyl indole and Acetyl indoles are regarded as very reactive olefins, which naturally undergo various addition reactions. The initial concept of using sulfonyl or acyl protection was already reported in ACS Comb. Sci. 2018, 20, 10, 573–578. So, the authors should justify the novelty of their approach.

Our responses:

We appreciated the critical comments by reviewer #1. YES, there are a number of reports regarding indole ring opening reactions. Most of these reports focused the transformation of indole ring to pyrazole or fused ring scaffold (Ref# 17, 41–43 in the revised manuscript). Similarly, as mentioned by reviewer #1, the study reported in ACS Comb. Sci. 2018, 20, 573–578 is also about the transformation of indole to pyrazole scaffold. After careful investigation, what we found is that the similarity of our study to that report by Manna and co-workers is only sharing the indole-based starting materials. Therefore, it is not the fair statement that our study is the simple addition reaction of sulfonyl (or acetyl) indole as reactive electrophiles. Actually, our work is fundamentally different from the work reported in forementioned ACS Comb. Sci. The following is the itemized point to show the difference between two studies including the scopes of the substrates.

Reviewer #1 claimed that “The initial concept of using sulfonyl or acyl protection was already reported in ACS Comb. Sci. 2018, 20, 573–578. So, the authors should justify the novelty of their approach.” In fact, the reaction reported in ACS Comb. Sci. referred by the reviewer #1 is compatible only with sulfonyl and acyl protected indole “regarded as very reactive olefins (stated by reviewer #1)”. For quick reference of editor, we attached few snap-shots from that article.

It is well documented that the carbonyl and sulfonyl groups form stable adducts with Lewis acids.³⁰ Hence, we judiciously introduced the carbonyl or sulfonyl containing directing groups like acyl, benzoyl and tosyl at the N1-position of indoles (page 573, left column)

Unfortunately, N1-methyl and N1-benzyl containing indole or unsubstituted (N1–H) indole failed to provide the corresponding pyrazole under these optimized reaction conditions (most of the starting materials were recovered). The reaction between 1- (page 575, right column)

However, our chemical transformation showed an excellent substrate scope which encompass the substituents on all possible positions around 3-formyl(aza)indoles along with a variety of N-protected 3-formyl(aza)indoles including free –NH, which is not possible in the reaction reported in ACS Comb. Sci. Our methodology also tolerates many functional groups, such as ester, amide, amine, hydroxyl, ether, even halide and -Bpin. We employed our transformation of (aza)indole-based natural products and drugs for late stage modifications. In this way, we demonstrated nearly 90 substrates by foreseeing the methodological significance and its applications owing to wide occurrence of highly privileged structures such as aniline, aminopyridine, and (aza)indole scaffolds in pharmaceuticals and natural products including many blockbuster drugs and W.H.O essential medicines.

Regarding the claim about use of N-acyl indoles, we are clarifying that the N-acetyl indoles are not compatible with our methodology and we didn't report such substrates in this manuscript. Under the reaction condition, the acetyl group undergoes the deprotection (Scheme S2). For this reason, the Nico-paracetamol/phenacetin were prepared by late-stage acetylation (Fig. 2b). In addition, as discussed earlier, we employed a wide range of N-modified/unmodified 3-formyl(aza)indoles, whereas Manna and co-workers should use N-sulfonyl/acyl indoles, not other indoles.

Therefore, our methodology is mechanistically different from the methodology developed by Manna and co-workers, that's why we have much wider substrate scope. Most importantly, our method does not depend on the electrophilicity of (aza)indoles, which is an essential part of Manna's report.

Comments from the reviewer #1

2. The authors also used Lewis acid like $Zn(OTf)_2$ instead of $BF_3 \cdot OEt_2$ because the difference in product yield was 3% only (which is within the error limit). $BF_3 \cdot OEt_2$ is much cheaper than $Zn(OTf)_2$, so the authors should justify why they have selected $Zn(OTf)_2$ over $BF_3 \cdot OEt_2$.

Our response:

We appreciate the interesting issue raised by reviewer #1. We do agree that $BF_3 \cdot Et_2O$ is less expensive than $Zn(OTf)_2$. Perhaps, as shown in the reaction optimization table, $ZnCl_2$ which is cheaper than $BF_3 \cdot Et_2O$ could be our preference for the reaction (Table S3, entry 18). However, our selection of $Zn(OTf)_2$ comes from the fact that $Zn(OTf)_2$ is easy to handle, less hygroscopic, and compatible to the other reactions at the high temperature (120 °C) in the laboratory scale. The yields (NMR and isolated) were also consistent with the use of $Zn(OTf)_2$ (Table S3, entry 19–21).

This skeletal transformation of (aza)indole bearing free -NH requires the condition with high temperature (120 °C), but $BF_3 \cdot Et_2O$ was found incompatible at this temperature, especially when we used ethanol as a solvent. In that case, we often observed the pressure build-up in the reaction vessel. Furthermore, in the practical aspect, the measuring of a precise amount of $BF_3 \cdot Et_2O$ (~2.0 μ L, 10 mol%) was a bit inaccurate due to its fuming nature, which might lead to the inconsistency of experimental outcomes. To avoid such manual errors, we have to use freshly prepared stock solution of $BF_3 \cdot Et_2O$ which cannot be applicable for a repeated usage. Most importantly, as shown in the optimization table (Table S4, entry 2), $BF_3 \cdot Et_2O$ is comparatively less efficient against indole scaffold, whereas $Zn(OTf)_2$ found compatible with both indole and azaindole scaffolds. That's why we selected $Zn(OTf)_2$ for the optimized reaction condition.

Entry	Catalyst (10 mol%)	Yield	Entry	Catalyst	%Yield ^b
1 ^a	-	56%	9	Ag_2CO_3	64%
2	CuI	63%	10	$AgNO_2$	66%
3	$CuCl_2 \cdot 2H_2O$	62%	11	$AgNO_3$	64%
4	$Cu(OTf)_2$	60%	12	$AlCl_3$	35%
5	$ZnCl_2$	71%	13	$FeCl_3$	65%
7	$Zn(OTf)_2$	66%	15	$BF_3 \cdot Et_2O$	63%
8	$AgOTf$	62%	16	CAN	71%
Deviation from Entry 1					
9	no catalyst, 80 °C				41%
10	no catalyst, NH_4OAc (5.0 equiv.)				65–70% (65%)
11	$ZnCl_2$ (10 mol%), NH_4OAc (3.0 equiv.)				59%
12	$ZnCl_2$ (5 mol%, 20 mol%)				70%
13	$Zn(OTf)_2$ (5 mol%, 15 mol%), NH_4OAc (5.0 equiv.)				70%
14	$BF_3 \cdot Et_2O$ (30 mol%)				74% (82%)
15	$BF_3 \cdot Et_2O$ (30 mol%), NH_4OAc (3.0 equiv.)				68%
16	$BF_3 \cdot Et_2O$ (30 mol%), NH_4OAc (5.0 equiv.)				74% (78%)
17 ^a	no catalyst, 120 °C				66%
18	$ZnCl_2$, 120 °C				72%
19 ^b	$Zn(OTf)_2$, 120 °C				72% (85%)
20	$Zn(OTf)_2$, NH_4OAc (5.0 equiv.), 120 °C				76%
21 ^b	$Zn(OTf)_2$, NH_4OAc (5.0 equiv.), 6 h				76% (90%)
22	no catalyst, 80 °C				41%
23	no catalyst, NH_4OAc (5.0 equiv.)				65–70% (65%)

Entry	Conditions	LC/MS SM:A:B	Yield ^d
1 ^b	No deviation		~79%
2	$BF_3 \cdot Et_2O$ (30 mol%)		~66%
3 ^c	1.5 equiv. of propiolate		(88%)
5 ^c	1.5 equiv. of propiolate, $Zn(OTf)_2$ (10 mol%)		(88%)
6	CH_3CN (2.0 mL) as solvent	33:60:7	
7	CH_3CN (2.0 mL), $BF_3 \cdot Et_2O$ (30 mol%)	6:92:2	
8	CH_3CN (2.0 mL), $Zn(OTf)_2$ (10 mol%)	11:87:2	75%
9 ^b	CH_3CN (2.0 mL), 120 °C	33:63:4	~82%
10	CH_3CN (2.0 mL), $BF_3 \cdot Et_2O$ (30 mol%), 120 °C	0:97:3	76%
11	CH_3CN (2.0 mL), $Zn(OTf)_2$ (10 mol%), 120 °C	0:97:3	99% (90%)
12	CH_3CN (2.0 mL), $Zn(OTf)_2$ (5 mol%), 120 °C		78%
13	CH_3CN (2.0 mL), $Zn(OTf)_2$ (10 mol%), 120 °C, 8–12h		75–80%

Comments from the reviewer #1

3. The authors should highlight the chemistry reported in ACS Comb. Sci. 2018, 20, 10, 573–578. The synthesis of ortho-unsubstituted nicotines from azaindole is an extension of the previous works.

Our response:

We do appreciate the work (reported in ACS Comb. Sci. 2018, 20, 573–578) which is a well-designed study to obtain pyrazole scaffolds from N-sulfonyl/acyl indoles. However, the points highlighted in that paper are quite different from those in this study where (aza)indole scaffolds were transformed to the vitamin B₃-substituted meta-aminobiaryl scaffolds via Aldol-type addition and intramolecular cyclization followed by C–N bond cleavage and re-aromatization. To support our argument, we have attached a few snap shots from ACS Comb. Sci. 2018, 20, 573–578, where the clear difference between two strategies can be seen.

(a) w.r.t the methodological outcome, the paper referred by reviewer #1 reported the synthesis of pyrazole scaffold using N-acyl/sulfonyl indoles and tosyl hydrazine. However, our report focuses on the robust synthesis of Vitamin B₃ fused with bioactive (hetero)arylamine scaffolds using 3-formyl (aza)indoles, NH₄OAc, and propiolate/propiolamide.

(abstract figure from ACS Comb. Sci. 2018, 20, 573–578)

(b) w.r.t the use of Lewis acid catalysts, the reaction reported in the aforementioned paper (referred by reviewer #1) works only in the presence of BF₃·OEt₂ (entry 1 and 2 in snap shot) and other Lewis and Bronsted acids were not compatible, especially Zn(OTf)₂ (entry 9 in snap shot). However, our reaction condition was optimized by using Zn(OTf)₂ as a suitable catalyst. Further, our reaction could yield the desired product even in the absence of Lewis acid catalyst.

ambient temperature (Table 1, entry 1). As expected, the model reaction in the absence of BF₃·OEt₂ failed to provide the desired product even at higher temperature (Table 1, entry 2). This (Page 574, left column, 2nd paragraph)

entry	acid/base (equiv)	solvent	time (h)	temperature (°C)	yield ^b (%)
1 ^c	BF ₃ ·OEt ₂ (0.3)	DCE	14	RT	55
2 ^d		DCE	48	RT → 50	ND ^e

(Page 574, right column, Table 1)

(Table 1, entry 4). Other Lewis and Bronsted acids, such as AlCl₃, FeCl₃, Zn(OTf)₂, I₂, AcOH, TsOH, TfOH, and TFA, were found to be ineffective (Table 1). Solvent screening was (Page 574, left column, 2nd paragraph)

7 ^d	AlCl ₃ (0.3)	DCE	14	50	ND
8 ^d	FeCl ₃ (0.3)	DCE	14	50	trace
9 ^d	Zn(OTf) ₂ (0.3)	DCE	14	50	ND

(Page 574, right column, Table 1)

(c) In the paper referred by reviewer #1, the reaction was not compatible with CH₃CN and alcoholic solvents. However, our reaction condition is compatible with CH₃CN/alcoholic solvents.

20 ^d	BF ₃ ·OEt ₂ (0.3)	CH ₃ CN	14	70	ND
21 ^d	BF ₃ ·OEt ₂ (0.3)	CH ₃ OH	14	50	ND

(Page 574, Table 1)

(d) The mechanistic pathway in the report referred by reviewer #1 is as described below

ABSTRACT: An unusual transformation of indoles to pyrazoles via an aromatic ring-opening strategy has been developed. The salient feature of this strategy involves the C2–N1 bond opening and concomitant cyclization reaction of the C2=C3 bond of the indole moiety with the tosylhydrazone, which proceeds under transition-metal and ligand free conditions. This ring-opening (Page 573, abstract)

complexation of Lewis acid (BF₃·OEt₂) with the oxygen of *N*-acyl/*N*-benzoyl/*N*-tosyl indoles sequestered the nitrogen lone-pair and allowing the activation of C2=C3 bond.^{10,45} The (Page 575, left column)

Scheme 2. Plausible Coupling Reaction of Activated Indoles with Tosylhydrazones

(Page 575, left column)

However, our strategy involves the transformation of (aza)indole scaffolds into substituted *meta*-aminoaryl scaffolds via Aldol-type addition, dehydration, and intramolecular cyclization followed by C–N bond cleavage and re-aromatization.

Reaction development inspired by Nature's Synthetic Strategy

Collectively, our methodology enables the late-stage conjugation of bioactive (hetero)arylamines with nicotines/nicotinamides and allows robust access with excellent functional group tolerance. This is our original work developed with the foresight of making an impact in drug discovery and ***our work is not the extension of any previous works.***

Comments from the reviewer #1

4. Authors suggest that C2-N bond cleavage is the driving force of the reaction. However, there is no significant proof for their claim. The authors should provide the proof for the formation of the intermediates. Only ¹H NMR based studies not enough for their claim. The provided ¹H NMR experiment does not completely support the given mechanism. We found the mechanism is not fully supporting the way authors have proposed.

Our response:

We do appreciate the thoughtful comment by reviewer #1, but we did not claim that “C2–N bond cleavage is the driving force of the reaction” either in our original manuscript or in the supplementary information. And we are confident that the NMR-based mechanistic studies using isotope-labeled substrates clearly elucidated the major issue regarding the reaction mechanism. This is as clearly recognized by reviewer #3 with the following statement “...The reaction mechanism has been clearly demonstrated by NMR studies, notably using deuterated substrates...”. We designed the substrates to obtain all possible deuterated products to reveal the reaction mechanism. Comparing the position of deuterium in the starting materials (**5f**, **5f'**, **5f''**, and **2a'**) with those in products clearly supports the pathway B initiated by Aldol-type addition than the pathway A initiated by imine formation followed by triene-based 6 π electrocyclization. Further, ¹H NMR spectra clearly support the formation of enamine from the ammonium acetate and ethyl propiolates (Scheme S4). We believe reviewer #2 also satisfied our mechanistic studies.

Comments from the reviewer #1

5. The in-situ generation of β -aminoacrylate from the corresponding alkyne in the presence of NH_4OAc is really a new concept?

Our response:

We appreciate this comment. To the best of our knowledge and as per data collected from advanced search engines using Scifinder[®] and Reaxys[®], we didn't find any report regarding the formation of β -aminoacrylates from the corresponding terminal alkynes in the presence of NH_4OAc . However, the method we adapted may not be a new concept, and the formation of enaminones was reported from internal ynones in the presence of NH_4OAc (*Asian J. Org. Chem.* **2018**, *7*, 1089–1092). As we described in the manuscript (with the support of literature precedence), we hypothesized *in situ* generation of β -aminoacrylate by activating the triple bond of terminal alkynes. This hypothesis was confirmed by the evidences collected by ¹H NMR experiments described in Scheme S4.

Comments from the reviewer #1

6. If this is really a novel approach, then why the authors have not mentioned about other nicotine derivatives except nicotinate only.

Our response:

I am afraid that I may not clearly understand the issue raised by reviewer #1. Nicotine is a potent alkaloid constituting pyridine-pyrrolidine scaffold. Nicotinate is the ester of nicotinic acid and they are not the derivatives of nicotine. Though there is a structural similarity between nicotine and nicotinate as m-substituted pyridine, they should be biologically different in their activities. If the reviewer questioned about other Vitamin B₃ analogues, we already reported the synthesis of nictinamide by using propiolamide in place of propiolates.

Comments from the reviewer #1

7. Reference to the opening of the C-N bond of indoles via the addition of two nucleophiles at the C2 positions should be mentioned such as J. Org. Chem. 1960, 486.

Our response:

We appreciate the thoughtful comment by reviewer #1. To address his/her comment, we incorporated this paper in the reference list of our revised manuscript (ref. #43).

Reviewer #2 (Remarks to the Author):

In this manuscript, Park and co-workers report development of a new strategy for the synthesis of meta-aminoaryl nicotinate from 2-formyl(aza)indoles. Unlike previous synthetic routes, this biomimetic method utilizes propiolates as enamine precursors and thus allows access to ortho-unsubstituted nicotinate. Additionally, this method enables the late-stage installation of bioactive (hetero)arylamines with nicotinate and nicotinamide. Most importantly, this process allows to produce a wide range of bisheterocyclic compounds that are difficult to access by the use of known methods, which is very useful for medchem studies and pharmaceutical industry. The authors also perform mechanistic studies in order to propose the mechanism for product formation. It's a good piece of work and I recommend its publication in Nature Communication after the following minor concerns are addressed.

Our response:

We deeply appreciate the reviewer #2 for his/her strong support on our work and kind consideration.

Comments from the reviewer #2

1) The authors should add one scheme to showed the methods of direct functionalization of C3 of pyridine, and briefly give a comparison with your method. (J. Am. Chem. Soc. 2011, 133, 6964; J. Am. Chem. Soc. 2011, 133, 16338)

Our response:

We deeply appreciate the insightful comment by reviewer #2. To address his/her comment, we incorporated the few sentences (Line 82–85, brief explanation of the direct functionalization of C3 of pyridine along with the comparison with our method) with the relevant scheme (Fig. 1f) in the revised manuscript (Line 82~85).

80 *via* aldol-type addition/dehydration and intramolecular cyclization followed by simultaneous re-
81 aromatization and C–N bond cleavage (Fig. 1e and Scheme S1).

82 **Actually, Pd-catalyzed highly regioselective direct C–H arylations at the C₃ and C₄ positions of**
83 **substituted pyridines are possible (Fig. 1f)^[39,40], only when pyridines contain electron-withdrawing groups (-**
84 **NO₂, -CN, -F, and -Cl) or directing group (carboximide). In addition, these methods cannot be employed to**
85 **conjugate functional aromatics like aryl amines to pyridines. Furthermore,** a retrosynthetic analysis of the
86 cross-coupling-based routes to *ortho*-nicotinated arylamines also revealed serious limitations, including long

The report from J. Am. Chem. Soc. 2011, 133, 16338 focused on Pd-catalyzed C–H arylation of pyridine at the C₃ and C₄ positions. This protocol is an excellent approach for direct arylation on pyridines only when pyridine contains electron-withdrawing groups. As stated by reviewer #2, this study can be an excellent comparison with our study, therefore we incorporated this paper as ref. #40 in the revised manuscript along with Fig. 1f. The report from J. Am. Chem. Soc. 2011, 133, 6964 concerned the Pd-catalyzed (ligand-promoted) C–H olefination at the C₃ position of pyridine. Even though this methodology is the direct functionalization of pyridine, but it is the C–H olefination of pyridine, not the C–H arylation. Instead, we found another relevant method reported by the same authors regarding the C–H arylation at the C₃ position of isonicotinic acid derivatives (Angew. Chem. Int. Ed. 2010, 49, 1275–1277), and added this report as ref. #39 in the revised manuscript. These methods are excellent approaches for direct C–H arylation of pyridines and good comparison with our study, but they are quite different from our skeletal transformation of (aza)indole scaffolds into Vitamin B₃-substituted meta-aminobiaryl scaffolds. After this change, our revised manuscript become more clear and informative.

(Figure 1f in the revised manuscript)

Comments from the reviewer #2

2) The authors should add 2-4 examples useful substituents such as -F, -CF₃.

Response:

We deeply appreciate the thoughtful comment by reviewer #2. To address his/her comment, we texted additional substrates containing -F and CF₃ substituents around all the possible positions of indole ring. As shown in 6o–6r and 6w–6y of Table 2 of the revised manuscript, we obtained the desired meta-aminobiaryl products containing -F and -CF₃ substituents. Due to the limited synthetic accessibility of C₇-CF₃-substituted indole, we didn't employ this example here. After this change, we can clearly show the robustness of this chemical transformation, even with free -NH (aza)indoles.

(Table 2 in the revised manuscript)

Comments from the reviewer #2

3) The authors should add additional results of 3-formylbenzofuran as the starting material.

Response:

We do appreciate this thoughtful suggestion by reviewer #2. As suggested, we clearly obtained Vitamin B₃-substituted meta-biaryl scaffolds from 3-formylbenzofuran using our optimized reaction condition. In this paper, however, we focused on the chemical transformation of (aza)indoles to meta-aminobiaryl scaffolds, and 3-formylbenzofuran may not fit nicely in this context. Actually, we are currently working on the follow-up study to access various substituents on nicotines as well as biaryl scaffolds, and we decided to include this additional results in the follow-up paper, not in this report, to maintain the consistency of this study. In fact, this study already contains a lot examples with ~250 pages of supporting information. I hope that reviewers understand our decision.

Compound: Ethyl 5-(2-hydroxyphenyl)nicotinate

 Pale yellow solid; Yield: 60% (29 mg); ^1H NMR (500 MHz, CDCl_3) δ 9.17 (d, $J = 1.5$ Hz, 1H), 8.99 (s, 1H), 8.52 (t, $J = 2.0$ Hz, 1H), 7.36–7.27 (m, 2H), 7.05 (td, $J = 7.5, 1.0$ Hz, 1H), 6.97 (d, $J = 8.0$ Hz, 1H), 6.30 (s, 1H), 4.44 (q, $J = 7.0$ Hz, 2H), 1.42 (t, $J = 7.2$ Hz, 3H); ^{13}C NMR (100 MHz, CDCl_3) δ 165.48, 153.75, 153.51, 148.56, 137.99, 134.47, 130.76, 130.31, 126.41, 123.97, 121.15, 116.60, 61.77, 14.43; IR: 2927, 1722, 1606, 1595, 1256, 1109, 750 cm^{-1} ; LRMS (ESI): m/z calcd for $\text{C}_{14}\text{H}_{14}\text{NO}_3$ $[\text{M}+\text{H}]^+$: 244.10; Found: 244.05.

Comments from the reviewer #2

4) Is the ester or amide on the alkyne 2 necessary for the reaction to take place? Author should add the results of the other electron-withdrawing group on the alkyne, for example: -Bpin, -CN.

Our response:

We appreciate very much for this insightful comment by reviewer #2, who is well-versed in the field and proposed additional experiments on the basis of his/her solid understanding of our late-stage chemical transformation. Based on reviewer #2's suggestion, we tested other types of terminal alkyne with electron-withdrawing groups to check whether they are suitable for this transformation. In the case of CN-substituted alkyne ($\equiv\text{C}-\text{CN}$), we cannot access this propiolonitrile due to its high reactivity and instability. Propiolonitrile can be synthesized by dehydration of propiolamide, but we didn't pursue that. In the case of Bpin-substituted alkyne ($\equiv\text{C}-\text{Bpin}$), the desired transformation under the optimized condition was not observed. Additionally, we tested ethynylsulfonylbenzene ($\equiv\text{C}-\text{SO}_2\text{Ph}$), we obtained the desired transformation to yield sulfonyl-substituted pyridine attached to meta-aminoaryl product (see the following Figure). Therefore, terminal alkynes with electron-withdrawing groups could be a suitable substrate in case-by-case.

However, we are currently working on the systematic modification of substituents on pyridine moiety (instead of nicotines) using β -keto-esters, sulfonates, and phosphonates. Using this reaction partners, we can access meta-amino(hetero)biaryl scaffolds with various substituents on pyridine moiety. We already obtained the desired transformation with R-substituted β -keto esters, sulfonates, and phosphonates, and we will search further with other β -keto nitrile or Bpin, etc. However, this content is out of the scope of this report, and we decide to focus on meta-amino(hetero)biaryl scaffolds conjugated with native nicotines. The new finding on systematic modification on general pyridine moiety will be reported as a follow-up paper in the near future.

Figure: The chemical transformation observed in our laboratory

Comments from the reviewer #2

5) In the supporting information, the authors should update the spectrum of product 5b-¹³C, 9a-¹³C, 6i-¹³C, 6d-¹³C, 6h-¹³C, 6j-¹³C/¹H, 10a-¹³C.

Our response:

Based on the reviewer #2's suggestion, we revised our supporting information with clear spectra of those products 5b-¹³C, 9a-¹³C, 6i-¹³C, 6d-¹³C, 6h-¹³C, 6j-¹³C/¹H, 10a-¹³C as well as other ¹H, ¹³C, and ¹⁹F NMR spectra. After these changes, our manuscript is much more informative and clear. We deeply appreciate the thoughtful comment by reviewer #2.

Reviewer #3 (Remarks to the Author):

The manuscript submitted by Park and co-workers constitutes an elegant and very complete piece of work in organic synthesis, allowing access to original molecules combining two motifs whose biological activity is recognized, a nicotinate motif and an aniline motif. The developed method consists of a three-component reaction between a 3-formylindole or azaindole, a beta-aminoacrylate and ammonium acetate, resulting in a domino sequence aldol-type addition / intramolecular cyclization / CN bond cleavage / re-aromatization. The products thus obtained are nicotinate or nicotinamide derivatives having a (hetero) arylamine substituent in the meta position. The substrates and reagents are simple and easily available, the reaction conditions are easy to implement, and the products are obtained with good to very good yields. The procedures and product characterizations are described in great detail in the supporting information, so that the experiments should be reproducible without difficulty.

Even if at first glance the sequence could be considered as a variant of the Bohlmann-Rahtz reaction, the originality lies in the use of indoles (or azaindoles) which rearrange themselves via the breaking of a C-N bond, releasing on the one hand the nicotinate or nicotinamide nucleus, and on the other hand the primary amine function of the aniline unit. This rearrangement is based on a clever working hypothesis which is inspired by the biosynthesis of vitamin B3 from L-Tryptophan. The reaction mechanism has been clearly demonstrated by NMR studies, notably using deuterated substrates.

The only downside that we could bring to this manuscript, considering in particular the strong experience of this team in the field of the development of new compounds for well-targeted biological applications, is the lack of discussion and arguments aimed at demonstrate the value of combining the two bioactive motifs concerned, for example by identifying a biological barrier for which these new products could provide solutions. Nevertheless, due to the nature of the two bioactive motifs thus conjugated, which revealed numerous and diverse biological activities, this work can reveal new bioactive compounds, which can be considered as derivatives of vitamin B3, by expanding the chemical space towards unexplored biological applications.

I support the publication of this article without any particular modification. Nevertheless, if we consider that the synthetic approach developed consists of a three-component reaction, other work leading to polysubstituted pyridines according to an MCR approach, including the formation of nicotinamide-type derivatives, could be cited in the references. We will retain for example the work of the Menendez team (ACS Comb. Sci. 2012, 14, 551) or those of the Rodriguez and Constantieux team (Chem. Commun. 2008, 4207 ; Chem. Eur. J. 2009, 15, 12945 ; Adv. Synth. Catal. 2012, 354, 2537).

Our response:

We deeply appreciate the comment by reviewer#3 with his/her comprehensive assessment of our work and for his/her strong support on our study. We agree with the reviewer's point of view about exploring the biological activities of conjugated bioactive compounds. We are currently pursuing the biological evaluation and the subsequent application of these novel compounds to various therapeutic area, and the outcome will be reported in the near future. But this paper focuses on the development of new methodology to access those new scaffolds.

Regarding the synthesis of polysubstituted pyridines, there are a number of interesting methodologies developed under the concept of MCR approach. Rodriguez and Constantieux team are a pioneer in this field and their contribution is quite exceptional. Actually, we already cited a review from Rodriguez and Constantieux team which is very comprehensive and covers the work of all forementioned citations. To address the comment by reviewer #3, we briefly described their contribution in our revised manuscript (MS; line 67–69) along with new citations (ref #35–38). After these changes, our manuscript is more informative and clear. Thanks for this thoughtful comments!

REVIEWERS' COMMENTS

Reviewer #1 (Remarks to the Author):

The author has satisfactorily answered all the question raised by this reviewer. Its a really great piece of work. Enjoined reading the modified manuscript.

Reviewer #2 (Remarks to the Author):

The scientific impact of this work meets the standards for publication and I strongly recommend this manuscript for publication without any particular modification